# LaMbDA: Local Latent Embedding Alignment for Cross-modal Time-Series Diffusion

## Abstract

We present a mutually aligned diffusion framework for cross-modal time-series generation that treats paired modalities $X$ and $Y$ as complementary observations of a shared latent dynamical process and couples their denoising trajectories through stepwise alignment of local latent embeddings. We instantiate this as LaMbDA (Local latent eMbedDing Alignment), a lightweight objective that enforces phase consistency by encouraging local latent neighborhoods of $X$ and $Y$ to inhabit a shared local manifold. LaMbDA augments the diffusion loss by incorporating first-order sequence-contrastive and second-order covariance alignment terms across modalities at matched timesteps. Aligning their local embeddings allows each modality to help denoise the other and resolve ambiguities throughout the reverse process. Human biomechanics provides a strong testbed for this approach: paired, synchronized measurements (e.g., joint kinematics and ground-reaction forces) capture the same movement state while reflecting practical constraints such as sensor dropout and cost. We evaluate LaMbDA extensively on biomechanical data and complement this with controlled studies on canonical synthetic dynamical systems (Lorenz attractor; double pendulum in non-chaotic and chaotic regimes) to probe generality under varying dynamical complexity. Across all these settings, aligning local latent statistics consistently improves generation fidelity, phase coherence, and representation quality for downstream probes, without architectural changes or inference overhead.

## 1 Introduction

Many real-world systems produce multiple data streams that are different views of the same evolving process Ren et al. (2022); Ashe & Briscoe (2006). These paired observations, measured by distinct sensors at different rates and with varying noise characteristics, often offer complementary perspectives of a shared underlying state Ren et al. (2022); Ashe & Briscoe (2006). Yet, they are rarely modeled jointly in a way that allows one stream to systematically disambiguate the other during generation and inference. Human movement is a representative example: joint kinematics, joint moments, and ground-reaction forces are synchronized, physically coupled measurements of a common locomotor state Winter (2009), but practical considerations (cost, setup complexity, occlusions, dropouts) often prevent observing all of them together or cleanly. A method that leverages complementarities between paired streams while remaining robust to partial, noisy observations would be broadly useful across such settings.

We introduce a mutually aligned cross-modal diffusion framework (Fig. 1) that treats two temporal modalities X and Y as complementary observations of a shared latent dynamical process. The core idea is to couple their reverse-diffusion trajectories by aligning local latent statistics at every diffusion step. Intuitively, local neighborhoods that correspond to the same phase should lie in nearby regions with similar variability. Enforcing this coherence allows each modality to help denoise and disambiguate the other throughout sampling. We instantiate this as LaMbDA (Local Latent Embedding Alignment), a lightweight objective that augments standard conditional diffusion without architectural changes or inference-time overhead. Two conditional models, $p_\theta(X \mid Y)$ and $p_\phi(Y \mid X)$, are trained in parallel, and at each diffusion step, they produce local embeddings from their noisy inputs. LaMbDA then encourages temporally matched windows to agree in their local statistics, while standard denoising losses train each model to reconstruct its target sequence. LaMbDA includes a first-order sequence-contrastive loss and a second-order covariance alignment

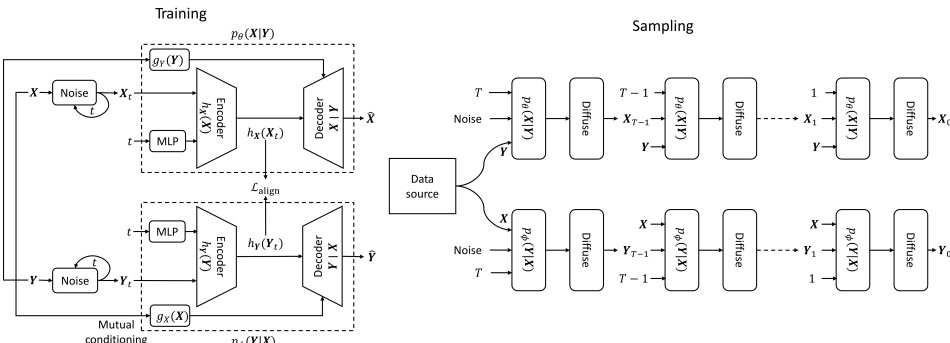

Figure 1: (Left) Mutually aligned cross-modal diffusion with latent embedding alignment. Diffusion processes, $p_\theta(\mathbf{X}|\mathbf{Y})$ and $p_\phi(\mathbf{Y}|\mathbf{X})$, generate data for modalities, $\mathbf{X}$ and $\mathbf{Y}$, respectively, guided by a condition derived from the other modality. During training, the latent representations, $h_X(\mathbf{X}_t, t)$ and $h_Y(\mathbf{Y}_t, t)$, of the two models are aligned using a local latent embedding alignment (LaMbDA) objective. Additionally, denoising and energy conservation objectives are applied to each modality's generated samples, $\hat{\mathbf{X}}$ and $\hat{\mathbf{Y}}$. During sampling, the model for each modality diffuses a noise signal across $T$ timesteps, guided by a condition from the other modality to generate samples of a given modality that temporally corresponds to the guiding signal.

loss between the modalities' latent spaces, ensuring that local neighborhoods not only match in their immediate representations but also preserve consistent internal correlation structure. The procedure is motivated by a dynamical-systems view: if both modalities are generated by the same latent state, then temporally matched windows from the two observation streams are related by a smooth change of coordinates Sauer et al. (1991).

We used human biomechanics as the primary testbed to evaluate this approach: paired, synchronized signals with clear physical coupling (angles, moments, forces) make it straightforward to verify whether cross-modal generation respects known relationships. The setting also reflects realistic sensing constraints—forces and moments are informative but expensive to measure, while kinematics are more accessible yet ambiguous—making biomechanics a stringent, informative setting for evaluating cross-modal diffusion under practical conditions. To probe generality to other dynamical systems and to study behavior under controlled dynamical complexity, we complement the real-world experiments with three canonical synthetic systems: the Lorenz attractor and the double pendulum in non-chaotic and chaotic regimes.

To the best of our knowledge, this is the first study to demonstrate cross-modal diffusion with latent alignment grounded in a dynamical systems perspective, and the first to showcase it in a biomechanical time series context. Our key contributions are: (1) We introduce a mutually aligned diffusion framework for cross-modal biomechanics synthesis through latent representation alignment. (2) We propose LaMbDA (Local latent eMBedDing Alignment), a lightweight, architecture-agnostic objective grounded in dynamical systems principles for aligning the latent representations of the modalities. (3) We demonstrate through experiments on real-world datasets and canonical dynamical systems that this simple latent alignment objective not only enhances generative quality but also maintains robust representations for downstream discriminative tasks.

## 2 RELATED WORK

**Diffusion models.** Denoising diffusion probabilistic models (DDPMs) have emerged as a powerful paradigm for generative modelling in high-dimensional settings. The seminal formulation introduced by Sohl-Dickstein et al. (2015) was refined by Ho et al. (2020), culminating in the state-of-the-art image synthesis performance Dhariwal & Nichol (2021). Beyond vision, subsequent adaptations to sequential data Yuan & Qiao (2024); Shen & Kwok (2023) have achieved competitive results in speech generation Kong et al. (2020); Chen et al. (2020a), time-series forecasting Kollovieh et al. (2024), and anomaly detection Xiao et al. (2023). Nevertheless, applications that employ diffusion models for *cross-modal* generation of time-series observations remain scarce.

**Cross-modal learning.** Cross-modal learning enables synthesis or interpretation in one modality using another, leveraging the complementarity of heterogeneous data (e.g., text, images, audio, video). Landmark systems like DALL·E Ramesh et al. (2021; 2022) and CLIP Radford et al. (2021) highlight the power of large-scale multimodal pretraining for generating coherent visuals from text. This paradigm now extends to music-to-dance Tseng et al. (2023); Zhuang et al. (2022), text-to-video Blattmann et al. (2023), text-to-motion Tevet et al. (2023), and audio-visual scene understanding Alamri et al. (2019). However, most approaches focus on *unidirectional* mappings (e.g., text → image), neglecting the inherently bidirectional nature of many real-world relationships. Moreover, cross-modal methods for physiological time-series, such as biomechanical signals, remain underexplored. These domains require models that capture continuous temporal signals and preserve cross-modal dynamics, highlighting the need for bidirectional cross-modal approaches tailored to time-series data.

**Representation alignment.** Representation alignment embeds heterogeneous inputs into a shared latent space that preserves structural and semantic content. Self-supervised methods like SimCLR Chen et al. (2020b), Barlow Twins Zbontar et al. (2021), and VICReg Bardes et al. (2021) have advanced unimodal pretraining for downstream tasks. However, these techniques are typically employed as auxiliary objectives rather than being integrated into diffusion-based generative frameworks, nor are they tailored for capturing temporal dependencies and inter-modal correlations in time-series data. In contrast, our approach employs latent alignment from a dynamical-systems perspective to enable bidirectional cross-modal generation of biomechanical time series.

**Biomechanical motion analysis and synthesis.** Biomechanical motion analysis combines kinematic data (e.g., joint angles) with kinetic data (e.g., ground reaction forces or GRFs). Foundational work by Winter emphasizes the interplay between these modalities in locomotion Winter (2009). Recent learning-based models have improved motion estimation Halilaj et al. (2018); Gurchiek et al. (2019); Horst et al. (2023), though they often rely on handcrafted features and unimodal inputs, limiting generalization. Multimodal fusion, such as combining motion capture with EMG, enhances muscle force and joint dynamics estimation Sartori et al. (2012); Young et al. (2014). Yet, cross-modal synthesis of biomechanical patterns remains largely unexplored, restricting adaptability across scenarios. We address this gap with a cross-modal generation method for biomechanical time series, grounded in latent representation alignment and dynamical systems principles. While broadly applicable, we focus on biomechanics, where modalities like joint angles, moments, and GRFs share an underlying dynamical structure that our approach exploits for robust generation.

## 3 CROSS-MODAL DENOISING DIFFUSION WITH LATENT ALIGNMENT

### 3.1 PROBLEM FORMULATION

Let $\{(\mathbf{X}_i, \mathbf{Y}_i)\}_{i=1}^N$ denote a paired dataset of time series trajectories of two modalities: joint kinematics and joint kinetics or variations thereof. Each trajectory $\mathbf{X}_i \in \mathbb{R}^{L \times d_X}$ and $\mathbf{Y}_i \in \mathbb{R}^{L \times d_Y}$ is a sequence of length $L$, with dimensions $d_X$ and $d_Y$ respectively. Our goal is to learn two generative models $p_\theta(\mathbf{X} \mid \mathbf{Y})$ and $p_\phi(\mathbf{Y} \mid \mathbf{X})$ ($\theta$ and $\phi$ are model parameters) such that one modality can be generated or reconstructed at full temporal resolution conditioned on the other.

### 3.2 DENOISING DIFFUSION

We adopt a denoising diffusion framework to learn these cross-modal distributions. Let $\beta_t$ for $t = 1, \ldots, T$ define a noise schedule that controls the noise variance at each step $t$ of the diffusion process. We define the following forward-noising processes for each modality:

$$\mathbf{X}_t = \sqrt{\beta_t}\,\mathbf{X}_{t-1} + \sqrt{1 - \beta_t}\,\boldsymbol{\epsilon}, \quad \mathbf{Y}_t = \sqrt{\beta_t}\,\mathbf{Y}_{t-1} + \sqrt{1 - \beta_t}\,\boldsymbol{\epsilon}, \tag{1}$$

where $\boldsymbol{\epsilon} \sim \mathcal{N}(0, \mathbf{I})$ is standard Gaussian noise.

We model the reverse process using conditional denoising diffusion processes, which predict the clean signal based on the noisy sample at each time step, $t$, and a condition derived from the other modality:

$$p_\theta(\mathbf{X}_0 \mid \mathbf{X}_t, g_Y(\mathbf{Y}), t), \quad p_\phi(\mathbf{Y}_0 \mid \mathbf{Y}_t, g_X(\mathbf{X}), t), \tag{2}$$

where $\theta$ and $\phi$ are parameters of the diffusion models, and $g_X(\cdot)$, $g_Y(\cdot)$ denote condition embedding functions for the modalities $\mathbf{X}$ and $\mathbf{Y}$, respectively.

We incorporate a mutual conditioning mechanism such that the generation of one modality is guided by the latent or encoded features from the other modality. Concretely, this means each decoder attends to both the noisy embedding of its own modality at time $t$ and a learned condition embedding derived from the other modality. For learning robust cross-modal representations, we enforce an alignment of the latent representations of the two modalities at each diffusion step. Since our modalities represent time-series data, we propose a modified alignment to ensure the temporal correlation of the local dynamics of the two modalities.

## 3.3 LATENT ALIGNMENT WITH DIFFUSION

**Dynamical systems background.**   In biomechanics, two modalities: joint kinematics and joint kinetics can be seen as *observational streams* of the *same underlying dynamical system* since they stem from the same musculoskeletal control process. Formally, consider a (possibly high-dimensional) hidden state, $\mathbf{Z} \in \mathbb{R}^{L \times d_Z}$ evolving according to an unknown dynamics:

$$\mathbf{Z}_{k+1} = f(\mathbf{Z}_k) + \boldsymbol{\eta}_k,$$

where $\boldsymbol{\eta}_k$ is a noise term. The observation functions, $o_\mathbf{X}$ and $o_\mathbf{Y}$, map the latent state into each modality's domain:

$$\mathbf{X}_k = o_\mathbf{X}(\mathbf{Z}_k), \quad \mathbf{Y}_k = o_\mathbf{Y}(\mathbf{Z}_k).$$

Under this perspective, $\mathbf{X}_k$ and $\mathbf{Y}_k$ arise from the same $\mathbf{Z}_k$ and thus should lie on correlated sub-manifolds of the global dynamical system. From Takens' embedding theorem Takens (2006) and related results in nonlinear time-series analysis Sauer et al. (1991), such partial views can still reconstruct consistent attractors or trajectories in phase space if appropriately embedded. This perspective underlies the motivation for aligning $\mathbf{X}$-space and $\mathbf{Y}$-space: if they come from the same dynamical manifold, then local segments of the latent dynamics should describe *the same underlying phase* and *the same local trajectories* (up to a smooth invertible transform).

**Local latent embedding alignment (LaMbDA).**   In our *mutually-aligned diffusion* approach, we train the diffusion models $p_\theta(\mathbf{X} \mid \mathbf{Y})$ and $p_\phi(\mathbf{Y} \mid \mathbf{X})$ simultaneously to reconstruct the two modalities, $\mathbf{X}$ and $\mathbf{Y}$, conditioned on each other. At each timestep $t$, the diffusion models produce latent embeddings, $\mathbf{Z}_{\mathbf{X},t} \in \mathbb{R}^{L \times d_Z}$ and $\mathbf{Z}_{\mathbf{Y},t} \in \mathbb{R}^{L \times d_Z}$. From a dynamical systems perspective, we may consider these latent embeddings as a reconstruction of the local phase space of the underlying dynamical system from each sensor's noisy observations. Since $\mathbf{Z}_{\mathbf{X},t}$ and $\mathbf{Z}_{\mathbf{Y},t}$ are reconstructions of the same underlying trajectory $\mathbf{Z}$, they should be aligned to each other.

We partition the latent sequences from the two models, $\mathbf{Z}_\mathbf{X}$ and $\mathbf{Z}_\mathbf{Y}$ into $P$ subsequences of length $S$,

$$\mathbf{Z}_\mathbf{X} = h_\mathbf{X}(\mathbf{X}) = [\mathbf{Z}_\mathbf{X}^{(1)}, \mathbf{Z}_\mathbf{X}^{(2)}, .., \mathbf{Z}_\mathbf{X}^{(P)}], \quad \mathbf{Z}_\mathbf{Y} = h_\mathbf{Y}(\mathbf{Y}) = [\mathbf{Z}_\mathbf{Y}^{(1)}, \mathbf{Z}_\mathbf{Y}^{(2)}, .., \mathbf{Z}_\mathbf{Y}^{(P)}], \quad (3)$$

where $P \approx L/S$. For each index $p = 1, \ldots, P$, $\mathbf{Z}_\mathbf{X}^{(p)}$ and $\mathbf{Z}_\mathbf{Y}^{(p)}$ represent short *temporally coherent* windows presumed to correspond to the *same* local dynamics. To encourage local manifold consistency, we propose a unified *local latent embedding alignment* (LaMbDA) objective that enforces both *first-order* and *second-order* consistency in the latent space.

**First-order (sequence-contrastive) alignment.**   To align the latent representations of corresponding time windows, we adopt a contrastive loss Oord et al. (2018) adapted to the temporal structure of the latent space by *pulling* together time-matched local latent subsequences, $(\mathbf{Z}_\mathbf{X}^{(p)}, \mathbf{Z}_\mathbf{Y}^{(p)})$, from the two modalities and *pushing* apart time-mismatched pairs from the same sequence, $(\mathbf{Z}_\mathbf{X}^{(p)}, \mathbf{Z}_\mathbf{Y}^{(q)}) \forall q \neq p$, as well as pairs from different sequences in a batch. Formally, for $P$ windows, we define:

$$\mathcal{L}_{\text{contrast}} = -\frac{1}{P} \sum_{p=1}^{P} \log \frac{\exp\big(\text{sim}(\mathbf{Z}_\mathbf{X}^{(p)}, \mathbf{Z}_\mathbf{Y}^{(p)})/\tau\big)}{\sum_q \exp\big(\text{sim}(\mathbf{Z}_\mathbf{X}^{(p)}, \mathbf{Z}_\mathbf{Y}^{(q)})/\tau\big) + \sum_{\text{other seq}}(\cdot)}, \quad (4)$$

where $\mathrm{sim}(.)$ represents a similarity function such as dot product or cosine similarity and $\tau$ is a temperature parameter. By locally aligning short-term dynamics, the model ensures that the local neighborhoods in the latent spaces derived from the two modalities reflect the same underlying state in each window.

**Second-order (covariance) alignment.** Beyond the pairwise similarity of the local latent manifold, we also align their internal structure using a covariance alignment term that enforces that the observation streams exhibit similar second-order statistics in their latent space. For each time step $l$, let $\mathbf{Z}_{\mathbf{X}}^{(l)}$ and $\mathbf{Z}_{\mathbf{Y}}^{(l)}$ denote the corresponding latent vectors for the two modalities. We compute the covariance matrices of these vectors (in a local neighborhood or across the entire sequence) and match them via:

$$\mathcal{L}_{\mathrm{cov}} = \frac{1}{L} \sum_{l=1}^{L} \mathrm{MSE}\Big(\mathrm{cov}\big(\mathbf{Z}_{\mathbf{X}}^{(l)}\big),\ \mathrm{cov}\big(\mathbf{Z}_{\mathbf{Y}}^{(l)}\big)\Big). \tag{5}$$

By matching the covariance matrices of $\mathbf{Z}_{\mathbf{X}}$ and $\mathbf{Z}_{\mathbf{Y}}$, we encourage both views to represent the same local manifold shape and correlation structure among latent dimensions, preserving the system's fundamental coupling and synergy patterns.

Finally, we form a single local latent embedding alignment (LaMbDA) loss by combining these two alignment components:

$$\mathcal{L}_{\mathrm{LaMbDA}} = \mathcal{L}_{\mathrm{contrast}} + \mathcal{L}_{\mathrm{cov}}. \tag{6}$$

A theoretical justification of our latent alignment approach grounded in Taken's embedding theorem is provided in Appendix A.1 (Theorem A.1).

**Energy conservation loss.** To maintain the biomechanical plausibility of generated trajectories, we add an energy conservation term to the loss, which encourages consistency between the energy of the reconstructed signal, $\hat{\mathbf{X}}$, and the ground truth signal, $\mathbf{X}$, of the two modalities. The energy conservation loss is computed as Gao et al. (2023):

$$\mathcal{L}_{energy}^{\mathbf{X}} = \|E(\hat{\mathbf{X}}) - E(\mathbf{X})\|_2, \quad E(\mathbf{X}) = \tfrac{1}{2}\nabla_l \mathbf{X}^2, \tag{7}$$

where $E(.)$ represents an energy function.

**Overall learning objective.** In addition to the local latent alignment, we use standard denoising objectives for each modality. Let $\mathcal{L}_{\mathrm{denoise}}^{\mathbf{X}}(\theta)$ and $\mathcal{L}_{\mathrm{denoise}}^{\mathbf{Y}}(\phi)$ be the respective MSE losses for generating $\mathbf{X}_0$ and $\mathbf{Y}_0$ from their noisy versions.

The *joint* objective is:

$$\mathcal{L}(\theta, \phi, \alpha) = \mathcal{L}_{\mathrm{denoise}}^{\mathbf{X}}(\theta) + \mathcal{L}_{\mathrm{denoise}}^{\mathbf{Y}}(\phi) + \alpha\,\mathcal{L}_{\mathrm{LaMbDA}}(\theta, \phi) + \mathcal{L}_{\mathrm{energy}}^{\mathbf{X}}(\theta) + \mathcal{L}_{\mathrm{energy}}^{\mathbf{Y}}(\phi), \tag{8}$$

where $\alpha$ is a learned weighting coefficient for the local latent embedding alignment. This learning objective poses conditional synthesis and cross-modal alignment as a single end-to-end optimization problem. We outline the training procedure in Algorithm 1.

## 4 EXPERIMENTS

**Datasets.** The evaluation is conducted on different biomechanical modalities that capture complementary signals of the locomotor process Embry et al. (2018). The dataset spans a broad range of locomotor conditions across a continuum of gait tasks, comprising approximately 1,540,000 samples collected from ten subjects over 27 distinct locomotion profiles, with walking speeds from 0.8 to 1.2 m/s and inclines from -10° to +10° in 2.5° increments. It includes both steady-state locomotion and transitions between conditions. The dataset provides precise time-varying joint kinematics, joint kinetics, and ground-reaction forces, offering a diverse and realistic foundation for evaluating the proposed method and ensuring relevance and robustness within biomechanics. In addition to the real-world biomechanical data, we evaluate on synthetic datasets derived from canonical dynamical systems (Fig. 5 in Appendix A.2): the Lorenz attractor Lorenz (2017) and the double pendulum in

---

**Algorithm 1:** Training Mutually-Aligned Diffusion with Local Latent Embedding Alignment

---

**Input:** Paired datasets $(\mathbf{X}, \mathbf{Y})$, noise schedule $(\beta_t)_{t=1}^T$, alignment weight $\alpha$, batch size $B$,
        sequence length $L$, subsequence length $S$, learning rate schedulers $\lambda_\phi, \lambda_\theta, \lambda_\alpha$.
**Output:** Learned parameters $\theta, \phi$ for $p(\mathbf{X}|\mathbf{Y})$ and $p(\mathbf{Y}|\mathbf{X})$
**Initialize** $\theta, \phi$, alignment weight, $\alpha$, and optimizers (e.g., AdamW).
**for** *epoch* $= 1 \ldots N_{epochs}$ **do**
    **foreach** *batch* $(\mathbf{X}_0, \mathbf{Y}_0)$ *of size B* **do**
        1. Sample Noisy Inputs:
        Sample $t \sim \text{Uniform}\{1, \ldots, T\};$    $\boldsymbol{\epsilon}_X, \boldsymbol{\epsilon}_Y \sim \mathcal{N}(\mathbf{0}, \mathbf{I});$
        $\mathbf{X}_t \leftarrow \sqrt{\beta_t}\,\mathbf{X}_0 \; + \; \sqrt{1-\beta_t}\,\boldsymbol{\epsilon}_X, \quad \mathbf{Y}_t \leftarrow \sqrt{\beta_t}\,\mathbf{Y}_0 \; + \; \sqrt{1-\beta_t}\,\boldsymbol{\epsilon}_Y;$
        2. Predict Denoised Inputs:   $\hat{\mathbf{X}}_0 \leftarrow p_\theta(\mathbf{X}_t, \mathbf{Y}, t), \quad \hat{\mathbf{Y}}_0 \leftarrow p_\phi(\mathbf{Y}_t, \mathbf{X}, t).$
        3. Compute Denoising Objective:
        $\mathcal{L}_{\text{denoise},X} = \|\mathbf{X}_0 - \hat{\mathbf{X}}_0\|^2, \quad \mathcal{L}_{\text{denoise},Y} = \|\mathbf{Y}_0 - \hat{\mathbf{Y}}_0\|^2.$
        4. Compute Energy Conservation Objective:
        $\mathcal{L}_{\text{energy},X} = \|E(\mathbf{X}_0) - E(\hat{\mathbf{X}}_0)\|^2, \quad \mathcal{L}_{\text{energy},Y} = \|E(\mathbf{Y}_0) - E(\hat{\mathbf{Y}}_0)\|^2.$
        5. Extract Latents & Compute Alignment Loss: $\mathbf{Z}_X \leftarrow h_X(\mathbf{X}_t, t), \;\; \mathbf{Z}_Y \leftarrow h_Y(\mathbf{Y}_t, t).$
        *(Subdivide $\mathbf{Z}_X, \mathbf{Z}_Y$ into local subsequences, each of length $S$.)*
        $\mathcal{L}_{\text{LaMbDA}} = \mathcal{L}_{\text{contrast}}(\mathbf{Z}_X, \mathbf{Z}_Y) \; + \; \mathcal{L}_{\text{cov}}(\mathbf{Z}_X, \mathbf{Z}_Y).$
        6. Combine Objectives:
        $\mathcal{L}_{\text{total}}(\theta, \phi, \alpha) = \mathcal{L}_{\text{denoise},X} + \mathcal{L}_{\text{denoise},Y} \; + \; \alpha\,\mathcal{L}_{\text{LaMbDA}} \; + \; \mathcal{L}_{\text{energy},X} + \mathcal{L}_{\text{energy},Y}.$
        7. Update Parameters:
        $\theta \leftarrow \theta - \lambda_\theta \nabla_\theta \mathcal{L}_{\text{total}}, \phi \leftarrow \phi - \lambda_\phi \nabla_\phi \mathcal{L}_{\text{total}}, \alpha \leftarrow \alpha - \lambda_\alpha \nabla_\alpha \mathcal{L}_{\text{total}}.$
    **end**
**end**
**return** $\theta, \phi$

---

non-chaotic and chaotic regimes. These synthetic benchmarks explicitly test the generality of the method in recovering meaningful temporal correspondence between observation channels derived from a shared underlying dynamical system.

**Evaluation.** We evaluate our model on cross-modal biomechanical observations: joint angles–joint moments, joint moments–ground reaction forces (GRFs), and joint angles–GRFs, using time-varying joint and body kinematics, joint kinetics, and force-plate data (see Fig. 2 for an example, and see Appendix A.2 for synthetic modality definitions). The three biomechanical modalities differ substantially in dimensionality: joint angles are 15-dimensional ($\theta_{hip}^{x,y,z}, \theta_{knee}^{x,y,z}, \theta_{ankle}^{x,y,z}, \theta_{foot}^{x,y,z}, \theta_{pelvis}^{x,y,z}$), joint moments 9-dimensional ($\tau_{hip}^{x,y,z}, \tau_{knee}^{x,y,z}, \tau_{ankle}^{x,y,z}$), and ground-reaction forces 3-dimensional ($\text{GRF}^{x,y,z}$). LaMbDA naturally supports alignment across such heterogeneous observation spaces because the alignment is performed in the latent space of the diffusion encoders rather than on the raw inputs. This design allows the method to align modalities with widely varying dimensionalities. In our experiments, we used a latent dimension of 128 based on a hyperparameter sensitivity analysis (see Appendix A.7). We use temporal segments of length $L = 300$ with 50% overlap (96.67% for synthetic data). This window length was selected based on a hyperparameter sensitivity analysis (see Appendix A.7, tab. 7) corresponds to two continuous gait cycles, enabling the model to learn transitions between cycles. Each model variant is trained and tested under multiple train–test splits, leaving out different participants and task profiles at each iteration rather than using a predefined test set. The test subsets include approximately 32k observations from 27 task profiles, reflecting varied speeds and conditions from a new user not seen during training. To stabilize training, we apply an exponential moving average (EMA) of parameters across batches and report aggregate results across all splits.

**Metrics.** The cross-modal generation performance was quantified using metrics that assess point-wise fidelity, distribution-level realism, temporal structure, and representational richness. Point-wise fidelity is measured using *mean-squared error (MSE)* between generated data and ground-truth physical observations; for reconstructions, the ground-truth signal is the temporal counterpart of the conditioning data. Distribution-level realism is measured using *Fréchet Inception Distance (FID)* to quantify differences between distributions of generated and real trajectories (lower

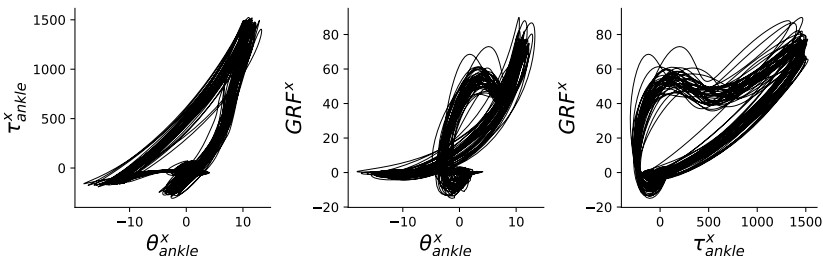

Figure 2: Raw trajectories showing the relationship between biomechanical modalities: joint angles–joint moments, joint moments–GRFs, and joint angles–GRFs, for an example locomotion task. The inherent periodicity in the relation shows that these modalities arise from a shared dynamical process and motivates our approach of latent alignment to discover the underlying dynamical system, which can then be used to generate trajectories for one modality conditioned on the other.

is better) Yu et al. (2021); Soloveitchik et al. (2021). Temporal structure is evaluated via the *predictive score*, computed as the forecasting error of a sequence prediction model when predicting future values of the ground truth sequences after training on generated data Yoon et al. (2019).

## 4.1 EFFECT OF LATENT ALIGNMENT ON CROSS-MODAL DIFFUSION

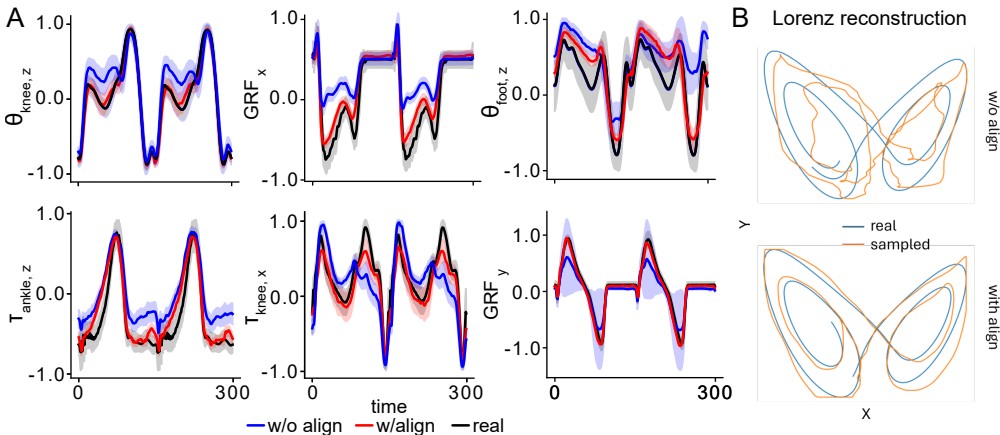

Figure 3: (A) Comparison of real and generated trajectories of (A) joint angles, moments, and GRF using models trained with and without latent alignment of diffusion models. The shaded region represents the standard deviation. Further visualizations are provided in Fig. 7 in Appendix A.4 . (B) Reconstruction of a window of 300 samples from the Lorenz attractor using models trained without and with alignment. Latent alignment improves the quality of generated samples in both cases.

We first tested our proposition that simply aligning the latent space of two independent conditioned diffusion models can improve cross-modal generation performance. For this, we analyzed whether the alignment of latent embeddings of the separate models $p_\theta(\mathbf{X}|\mathbf{Y})$ and $p_\phi(\mathbf{Y}|\mathbf{X})$ that learn to generate each modality can improve the quality of their generated gait trajectories using different metrics such as MSE, FID, and predictive score. We found that latent alignment through local latent embedding alignment (LaMbDA) improves the cross-modal generation accuracy for all the different modalities tested (Tab. 1). This was further illustrated by a better agreement of the trajectories generated by the aligned models with the ground truth trajectories (Fig. 3A).

It should be noted that the difference in the metrics for $\mathbf{X} \mid \mathbf{Y}$ and $\mathbf{X} \mid \mathbf{Y}$ arises from the inherent information assymetry in the underlying biomechanical modalities. Joint angles describe the configuration and motion of the limb segments, which provides a richer representation of the movement state than the downstream kinetic signals. Joint moments depend on these kinematics through inverse

| Modality pair | Direction | MSE ↓ | | FID ↓ | | Pred ↓ | |
|---|---|---|---|---|---|---|---|
| | | w/o align | w/ align | w/o align | w/ align | w/o align | w/ align |
| angles–moments | $X\|Y$ | 0.18±0.03 | **0.14±0.02** | 37.8±9.6 | **32.4±7.2** | 0.18±0.06 | **0.16±0.03** |
| | $Y\|X$ | 0.08±0.02 | **0.07±0.01** | 20.4±12.1 | **14.2±2.8** | 0.08±0.01 | **0.07±0.01** |
| angles–GRF | $X\|Y$ | 0.22±0.03 | **0.19±0.03** | 66.7±51.5 | **40.4±8.3** | 0.30±0.23 | **0.16±0.02** |
| | $Y\|X$ | 0.07±0.03 | **0.06±0.03** | 24.8±34.2 | **5.8±3.6** | 0.12±0.12 | **0.08±0.07** |
| moments–GRF | $X\|Y$ | 0.08±0.02 | **0.07±0.02** | 16.5±4.0 | **13.7±3.2** | 0.08±0.01 | **0.07±0.01** |
| | $Y\|X$ | 0.03±0.02 | **0.03±0.02** | 6.6±2.5 | **4.3±2.5** | 0.07±0.04 | **0.05±0.04** |

Table 1: Comparison of cross-modal generation performance (mean ± std) of the conditional diffusion models for each modality pair, trained with and without latent alignment. The performance is evaluated using the discrepancy (MSE) between generated and ground truth trajectories, Fréchet Inception Distance (FID), and predictive score (predictive error), all of whose lower values indicate better performance. Training with latent alignment improves cross-modal generation quality across all modalities tested under all the different metrics evaluated here.

dynamics, and GRFs depend on the global body motion and foot–ground interaction, which are also strongly constrained by the kinematic trajectory. As a result, the angle modality typically contains more upstream information about the ongoing movement than moments or GRFs. This makes reconstructing moments or GRFs conditioned on angles easier than reconstructing the full kinematic trajectory from the kinetic measurements. This observation is supported by our entropy analysis (Appendix A.8), which shows conditional entropy $H(\mathbf{X} \mid \mathbf{Y})$ to be consistently higher than $H(\mathbf{X} \mid \mathbf{Y})$ when $X$ is the joint angles (Tab. 8).

| Dynamical system | model | w/o alignment | with alignment |
|---|---|---|---|
| Lorenz attractor | $X\|Y$ | 0.678 | **0.425** |
| | $Y\|X$ | 0.135 | **0.004** |
| Double pendulum (non-chaotic) | $X\|Y$ | **2.5e-3** | **2.5e-3** |
| | $Y\|X$ | 6.6e-3 | **6.4e-3** |
| Double pendulum (chaotic) | $X\|Y$ | 0.042 | **0.028** |
| | $Y\|X$ | 0.031 | **0.021** |

Table 2: Comparison of cross-modal generation performance (quantified by MSE) of the conditional diffusion models trained with and without latent alignment on canonical dynamical systems.

**Additional experiments on synthetic data.** We found that the latent alignment strategy also improves model performance in trajectory reconstruction from different canonical dynamical systems (Tab. 2 and Fig. 3B). Performance improvement was more pronounced in the chaotic regime than in a non-chaotic regime (Tab. 2), underlying the ability of our approach to model complex dynamical relationships between modalities. These additional experiments on synthetic benchmarks establish LaMbDA as a powerful method for cross-modal synthesis when the two modalities originate from a shared underlying dynamical system.

## 4.2 COMPARISON WITH BENCHMARKS

We evaluated the quality of the latent representations learned by LaMbDA against state-of-the-art self-supervised alignment methods, such as SimCLR Chen et al. (2020b), Barlow Twins Zbontar et al. (2021), and VICReg Bardes et al. (2021), and a simple baseline that minimizes mean-squared error between the latents of the two models. The comparison used a downstream task: classification of the locomotion task label. Each input sample from either modality $X$ or $Y$ belongs to one of 27 locomotion tasks defined by walking speed and ground incline. A linear or non-linear classifier was trained on diffusion-encoder outputs to predict the task label. Higher linear/non-linear probing scores indicate better discrimination of locomotion tasks in latent space, hence higher representation quality. LaMbDA outperformed the state-of-the-art alignment methods on four of the six models and ranked second on the remaining two (Tab. 3).

To strengthen our empirical evaluation, we further benchmark LaMbDA against powerful cross-modal generative baselines. We evaluate two state-of-the-art cross-modal diffusion frameworks, CDCD Zhu et al. (2023) and CMMD Yang et al. (2024), as well as conditional generative models, including

| | Linear probing ↑ | | | | | |
|---|---|---|---|---|---|---|
| Modality pair | Angles – Moments | | Angles – GRF | | Moments – GRF | |
| Alignment | $X \mid Y$ | $Y \mid X$ | $X \mid Y$ | $Y \mid X$ | $X \mid Y$ | $Y \mid X$ |
| Barlow | 0.70±0.08 | 0.71±0.08 | 0.64±0.06 | 0.63±0.06 | 0.53±0.08 | 0.51±0.10 |
| SimCLR | 0.82±0.04 | **0.79±0.06** | 0.68±0.06 | **0.78±0.04** | 0.78±0.04 | 0.80±0.08 |
| MSE | 0.72±0.04 | 0.72±0.06 | 0.62±0.10 | 0.74±0.03 | **0.82±0.04** | **0.83±0.05** |
| VICReg | 0.65±0.09 | 0.64±0.06 | 0.66±0.07 | 0.62±0.07 | 0.54±0.07 | 0.59±0.09 |
| LaMbDA | **0.86±0.05** | 0.78±0.05 | **0.80±0.06** | 0.75±0.04 | 0.82±0.04 | 0.83±0.07 |
| | Nonlinear probing ↑ | | | | | |
| Barlow | 0.72±0.06 | 0.73±0.07 | 0.66±0.08 | 0.68±0.05 | 0.63±0.07 | 0.57±0.10 |
| SimCLR | 0.83±0.05 | **0.80±0.06** | 0.74±0.07 | **0.81±0.05** | 0.64±0.07 | 0.68±0.09 |
| MSE | 0.74±0.05 | 0.75±0.06 | 0.65±0.10 | 0.76±0.05 | **0.85±0.05** | **0.85±0.05** |
| VICReg | 0.64±0.09 | 0.64±0.07 | 0.72±0.07 | 0.66±0.06 | 0.64±0.07 | 0.68±0.09 |
| LaMbDA | **0.86±0.05** | **0.80±0.05** | **0.83±0.06** | 0.78±0.05 | **0.85±0.05** | 0.84±0.05 |

Table 3: Quality of learned representations of different latent alignment methods quantified as the performance on locomotion profile classification using linear and nonlinear probes (mean and standard deviation across test sets, bold indicates best performing and underline indicates second best performing). Local latent embedding alignment (LaMbDA) outperforms state-of-the-art self-supervised methods across four out of six modalities tested, and performed second best in the remaining two modalities.

a CVAE (with and without latent alignment) and a transformer-based time-series regressor (Tab. 4). Across these additional baselines, LaMbDA consistently achieves the strongest performance. While adding latent alignment improves CVAE performance, it still trails behind the diffusion-based alignment achieved by LaMbDA.

| | Linear probing | | Nonlinear probing | |
|---|---|---|---|---|
| $X$ = **Angles**, $Y$ = **Moments** | $X \mid Y$ | $Y \mid X$ | $X \mid Y$ | $Y \mid X$ |
| Transformer Regressor | 0.05 | 0.05 | 0.05 | 0.05 |
| CVAE w/o align | 0.48 | 0.35 | 0.65 | 0.4 |
| CVAE w/align | 0.54 | 0.53 | 0.69 | 0.67 |
| CDCD | 0.06 | 0.47 | 0.04 | 0.5 |
| CMMD | 0.65 | 0.78 | 0.76 | **0.89** |
| **LaMbDA (ours)** | **0.89** | **0.88** | **0.89** | 0.88 |

Table 4: Comparison of downstream task performance against a transformer regressor baseline, conditional generative baseline, CVAE with and without latent alignment, and cross-modal diffusion benchmarks such as CDCD, CMMD for one cross-validation iteration.

### 4.3 EFFECT OF ALIGNMENT ON THE LEARNED REPRESENTATIONS

Next, we evaluated how latent alignment influences representation quality. We first visualized the latent spaces of models trained without and with alignment using UMAP McInnes et al. (2018). With alignment, the two spaces were highly correlated, and same-task samples occupied overlapping subspaces (Fig. 4). This was further corroborated by superior downstream linear classification of locomotion task profiles. Thus, aligning the two latent spaces enhances the representational quality of individual modalities, likely by capturing shared or complementary information from the two views. This effect is not due to mutual conditioning at the decoder or the energy-conservation objective, since non-aligned models were also trained with these components.

### 4.4 ABLATIONS

Finally, we conducted an ablation study to assess the contribution of each component to our overall loss term (Eq. 8). Specifically, we removed the energy conservation objectives ($\mathcal{L}_{energy,\mathrm{X}}$ and $\mathcal{L}_{energy,\mathrm{Y}}$), the covariance alignment objective ($\mathcal{L}_{\mathrm{cov}}$), and the contrastive alignment objective

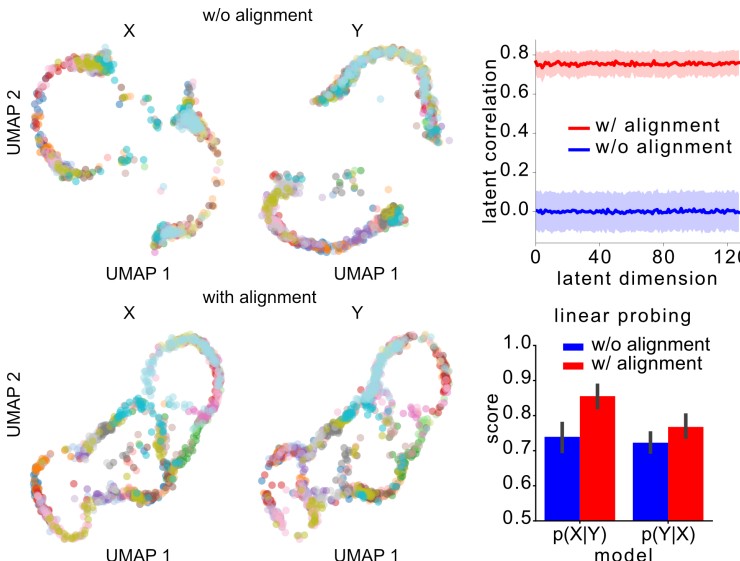

Figure 4: (Left) Latent embeddings of $p(X \mid Y)$ and $p(Y \mid X)$ on a held-out subject, trained with and without latent alignment (color-coded by locomotion task). Without alignment, representations show strong modality-specific separation, whereas with alignment, the two latent spaces merge, and same-task samples occupy overlapping subspaces. Also see Fig. 9 in Appendix A.4. (Right top) Correlation between the two modality-specific latent spaces on held-out test data (shaded area = standard deviation). Models trained with alignment exhibit high cross-modal correlation. (Right bottom) Linear-classifier performance for discriminating locomotion tasks from each modality-specific latent space (error bars = standard deviation). Alignment improves accuracy, indicating clearer task separation in the latent space.

| | $\mathcal{L}_{\text{energy}}$ | $\mathcal{L}_{\text{contrast}}$ | $\mathcal{L}_{\text{cov}}$ | $X\|Y$ | $Y\|X$ |
|---|:---:|:---:|:---:|---|---|
| LaMbDA w/o $\mathcal{L}_{\text{contrast}}$ | ✓ | | ✓ | 0.18±0.03 | 0.08±0.03 |
| LaMbDA w/o $\mathcal{L}_{\text{cov}}$ | ✓ | ✓ | | 0.17±0.03 | 0.07±0.02 |
| LaMbDA w/o $\mathcal{L}_{\text{energy}}$ | | ✓ | ✓ | 0.17±0.02 | 0.07±0.02 |
| **LaMbDA** | ✓ | ✓ | ✓ | **0.14±0.02** | **0.07±0.01** |

Table 5: Effect of ablation of individual components of the objective on the model performance measured using MSE (Mean and standard deviation across test sets; lower the better). Removing each component worsens the model's cross-modal generation capability, whereas all the components together are required to achieve the best performance.

($\mathcal{L}_{\text{constrast}}$), individually, and compared these variants against the full objective. Our results show that each component is necessary for achieving the best performance from our method (Tab. 5).

## 5 CONCLUSIONS

We presented a novel mutually-aligned diffusion framework for cross-modal biomechanical time-series generation, grounded in a dynamical systems perspective. By applying a local latent embedding alignment, comprising, first-order (sequence-contrastive) and second-order (covariance) alignment at each diffusion time step, our approach synthesizes realistic kinematic and kinetic trajectories, preserving biomechanically consistent relationships across the two modalities. Experiments show that this simple alignment strategy produces more accurate signal generation compared to baselines, and also enhances performance in downstream tasks, demonstrating its utility in both generative and discriminative contexts.

**Limitations.** LaMbDA assumes that the paired modalities arise from a shared latent dynamical process. Extending the method to settings where this assumption is violated is an important direction for future work. Furthermore, scaling the method beyond two modalities may require additional strategies such as centroid-based alignment or coordinated pairwise alignment.

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

# A APPENDIX

## A.1 THEORETICAL JUSTIFICATION OF LATENT ALIGNMENT

**Theorem A.1.** *Let $(M, \phi_t)$ be a compact $C^2$ dynamical system on a smooth manifold $M$ of dimension $d_Z$. Let $o_X : M \to \mathbb{R}^{d_X}$ and $o_Y : M \to \mathbb{R}^{d_Y}$ be smooth generic observation functions representing two different measurement modalities (e.g., kinematics and kinetics). Define the delay embedding operators:*

$$\mathcal{E}_X(Z) = \left[ o_X(Z), o_X(\phi_\tau(Z)), \ldots, o_X(\phi_{(\kappa-1)\tau}(Z)) \right] \in \mathbb{R}^{\kappa \times d_X},$$

$$\mathcal{E}_Y(Z) = \left[ o_Y(Z), o_Y(\phi_\tau(Z)), \ldots, o_Y(\phi_{(\kappa-1)\tau}(Z)) \right] \in \mathbb{R}^{\kappa \times d_Y},$$

*for some fixed delay $\tau > 0$ and embedding dimension $\kappa \in \mathbb{N}$.*

*If $\kappa d_X > 2d_Z$ and $\kappa d_Y > 2d_Z$, then for generic $o_X$ and $o_Y$, both $\mathcal{E}_X$ and $\mathcal{E}_Y$ are $C^1$ embeddings of $M$.*

*Consequently, their images $\mathcal{M}_X := \mathcal{E}_X(M)$ and $\mathcal{M}_Y := \mathcal{E}_Y(M)$ are diffeomorphic to $M$ and thus to each other. In particular, the map*

$$\Psi := \mathcal{E}_Y \circ \mathcal{E}_X^{-1} : \mathcal{M}_X \to \mathcal{M}_Y$$

*is a diffeomorphism.*

*Proof.* This follows directly from the generalized Takens' embedding theorem for vector-valued observations Sauer et al. (1991). Since $M$ is compact and the flows $\phi_t$ are smooth, the compositions $o_X \circ \phi_t$ and $o_Y \circ \phi_t$ remain $C^2$ functions. Under the assumption that $\kappa d_X > 2d_A$ and that $o_X$ is a generic smooth map, the embedding $\mathcal{E}_X : M \to \mathbb{R}^{\kappa \times d_X}$ is an injective immersion and hence an embedding. The same holds for $\mathcal{E}_Y$.

Because both embeddings are diffeomorphisms from $M$ to their respective images $\mathcal{M}_X$ and $\mathcal{M}_Y$, their composition $\Psi := \mathcal{E}_Y \circ \mathcal{E}_X^{-1}$ is a smooth bijection with a smooth inverse—i.e., a diffeomorphism between $\mathcal{M}_X$ and $\mathcal{M}_Y$. $\qquad\square$

**Implication for Local Alignment.** In practice, we assume that the diffusion model encoders learn latent representations $Z_X^{(i)} \approx \mathcal{E}_X(Z_i)$ and $Z_Y^{(i)} \approx \mathcal{E}_Y(Z_i)$ from local trajectory windows. The diffeomorphism $\Psi$ implies that

$$Z_Y^{(i)} = \Psi(Z_X^{(i)}),$$

and under smoothness of $\Psi$, we can locally approximate it by a first-order Taylor expansion:

$$Z_Y^{(i)} \approx A_i Z_X^{(i)} + b_i,$$

where $A_i = J_\Psi(Z_X^{(i)})$ is the Jacobian. Thus, minimizing both $\|Z_X^{(i)} - Z_Y^{(i)}\|^2$ (first-order alignment) and $\left\| \mathrm{Cov}(Z_X^{(i)}) - \mathrm{Cov}(Z_Y^{(i)}) \right\|_F^2$ (second-order alignment) encourages local linear agreement of $\Psi$, which our Local Latent Embedding Alignment (LaMbDA) loss is designed to enforce.

## A.2 SYNTHETIC BENCHMARKS

### A.2.1 LORENZ ATTRACTOR

The Lorenz system is a three-dimensional continuous-time dynamical system defined by:

$$\begin{aligned} \dot{x} &= \sigma(y - x), \\ \dot{y} &= x(\rho - z) - y, \\ \dot{z} &= xy - \beta z, \end{aligned} \tag{9}$$

where $x, y, z \in \mathbb{R}$ represent the state variables and we use the canonical chaotic parameters $\sigma = 10$, $\rho = 28$, and $\beta = 8/3$. This system is known for its sensitive dependence on initial conditions and its characteristic "butterfly"-shaped strange attractor (Fig. 5A).

We integrate the system using the Runge–Kutta 4th order method (RK4) with a timestep $\Delta t = 0.01$, starting from the initial condition $\mathbf{x}_0 = [5.0, 5.0, 5.0]$ for 10000 steps. To construct a multimodal

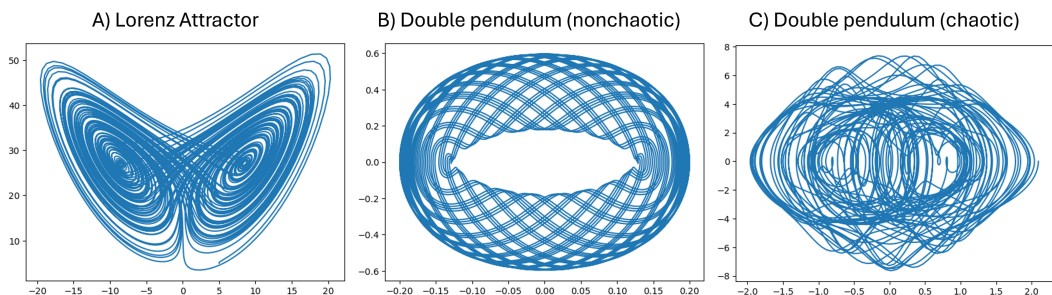

Figure 5: Phase-space visualizations of attractors from the synthetic dynamical systems used in our evaluation. (A) Lorenz attractor ($x - z$ projection), exhibiting classical chaotic structure. (B) Double pendulum in a non-chaotic regime, forming a smooth toroidal attractor. (C) Double pendulum in a mildly chaotic regime, producing a distorted, non-periodic attractor. These attractors illustrate the diversity of dynamical complexity used to test cross-modal alignment.

setting, we define the scalar time series $x(t)$ as **modality 1** and $z(t)$ as **modality 2**. The models trained with local latent embedding alignment (LaMbDA) reconstructed the dynamical systems trajectories more precisely than those trained without alignment (Fig. 6).

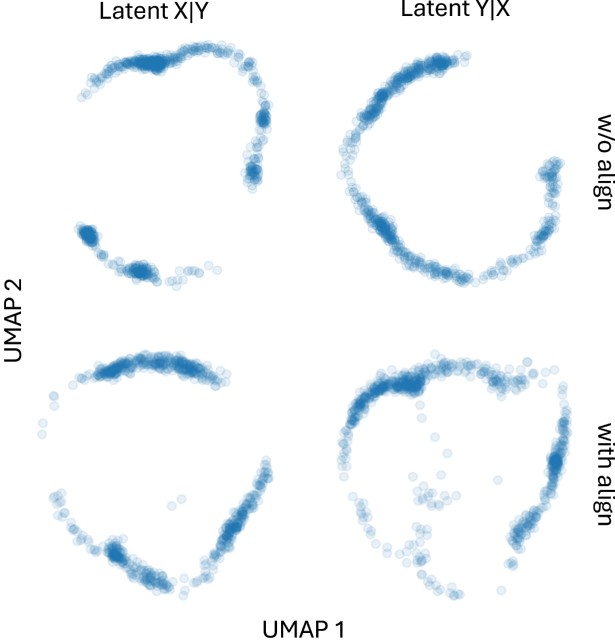

Figure 6: UMAP visualization of latent spaces of the cross-modal diffusion models trained on Lorenz attractor data without (top) and with (bottom) latent alignment. The latent embedding of the models trained with alignment shows a higher correlation.

### A.2.2 DOUBLE PENDULUM

The double pendulum consists of two rigid links swinging in a vertical plane, with angular displacements $\theta_1(t)$ and $\theta_2(t)$, and corresponding angular velocities $\omega_1(t) = \dot{\theta}_1$ and $\omega_2(t) = \dot{\theta}_2$. The system evolves according to a set of coupled second-order nonlinear differential equations derived from a Lagrangian formulation. We consider a simplified configuration with unit-length rods, equal masses

$m_1 = m_2 = 1$, and gravitational acceleration $g = 9.81$ m/s$^2$. Integration is performed using a fourth-order Runge–Kutta (RK4) method with timestep $\Delta t = 0.01$ for 10000 steps.

To evaluate alignment under varying dynamical conditions, we simulate the double pendulum under two distinct regimes:

- **Non-chaotic regime:** We initialize the system with small angular displacements and zero initial velocities: $\theta_1 = 0.2$, $\theta_2 = 0.18$, $\omega_1 = \omega_2 = 0$. This results in quasi-periodic motion with smooth, stable trajectories, ideal for baseline alignment under low dynamical complexity (Fig. 5B).

- **Chaotic regime:** We use higher initial energy by setting $\theta_1 = 2.1$, $\theta_2 = -0.1$, $\omega_1 = \omega_2 = 0$. This produces irregular, aperiodic motion characteristic of weakly chaotic behavior, while remaining numerically stable over long simulation horizons (Fig. 5C).

In both settings, we construct a multimodal observation setup where **modality 1** is the scalar angular velocity $\omega_1(t)$ of the first link, and **modality 2** is $\omega_2(t)$, the angular velocity of the second link.

## A.3 BROADER IMPACTS

The proposed framework for mutually aligned cross-modal diffusion opens a wide range of possibilities in scenarios where one or more data streams are missing, noisy, or difficult to measure directly. In wearable assistive devices and robotics, it can infer absent or corrupted sensor inputs, such as force or torque data from more accessible modalities, thereby enhancing real-time control and reliability despite equipment constraints or sensor failure. Within the biomechanical domain, the ability to simulate perturbations in one modality and observe their repercussions in another offers powerful insights into how different aspects of locomotion co-evolve, informing the design of targeted rehabilitation protocols and sophisticated training regimens. By allowing for more efficient sensor setups, the framework supports clinicians and researchers in long-term monitoring without requiring extensive instrumentation, broadening the potential for in-home rehabilitation and remote athlete performance tracking. Beyond biomechanics, the fundamental principles behind our cross-modal diffusion paradigm can be extended to other domains where interacting data streams arise from a shared dynamical process. For instance, in climate modeling, it could align or impute different types of geospatial and atmospheric measurements to refine weather or environmental forecasts. Even financial modeling could benefit from aligning time-series of economic indicators or market signals to better predict systemic interactions. Importantly, our experiments on synthetic dynamical systems (e.g., Lorenz attractor and double pendulum) demonstrate that the proposed LaMbDA framework yields similar improvements in reconstruction accuracy and latent structure alignment, highlighting the generality and applicability of our method across domains governed by shared latent dynamics.

**Ethical Considerations.** The ability to reconstruct missing data from alternative sources raises important questions about privacy, consent, and fairness, particularly when dealing with sensitive physiological information. These concerns underscore the need for robust regulatory frameworks and ethical practices to ensure responsible research and real-world implementations.

## A.4 FURTHER VISUALIZATIONS

We provide further visualizations of reconstructed trajectories, failure cases and latent space visualizations in Fig. 7–9.

## A.5 SHARED ENCODER ARCHITECTURE

Another method to derive a shared latent space for the two modalities (kinematics and kinetics) is to use a single encoder which takes both the modalities together as inputs. For cross-modal generation, one can use modality-specific decoder heads which takes the other modality as conditioning input. Although this method eliminates the need for explicit alignment in the latent space (and uses only the denoising objective), we found that the generation quality is inferior compared to the modality-specific encoder-decoder architecture that we propose.

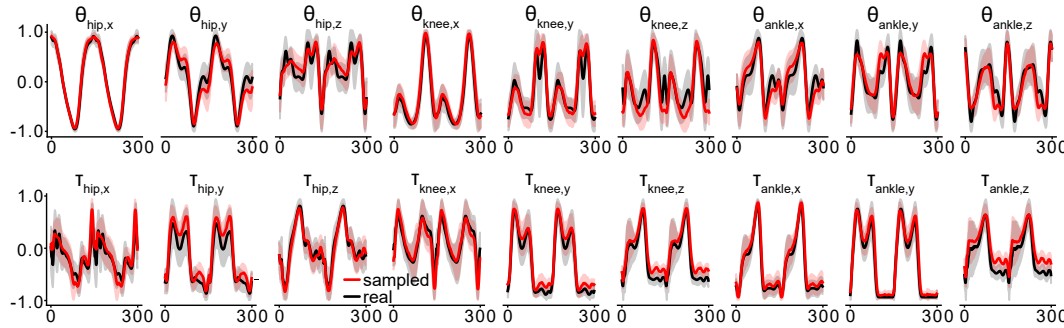

Figure 7: Real (black) and sampled (red) trajectories of joint angles (top) and joint moments (bottom) generated by latent aligned cross-modal diffusion models. All the generated trajectories follow the ground truth trajectories closely. Shaded region represents standard deviation.

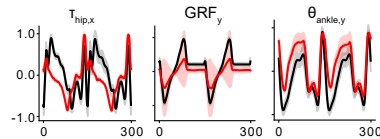

Figure 8: Example failure cases of the model for the prediction of the three modalities. Failure cases mostly occur when the underlying true signal shows high variability, or due to sign changes in the sampled signals.

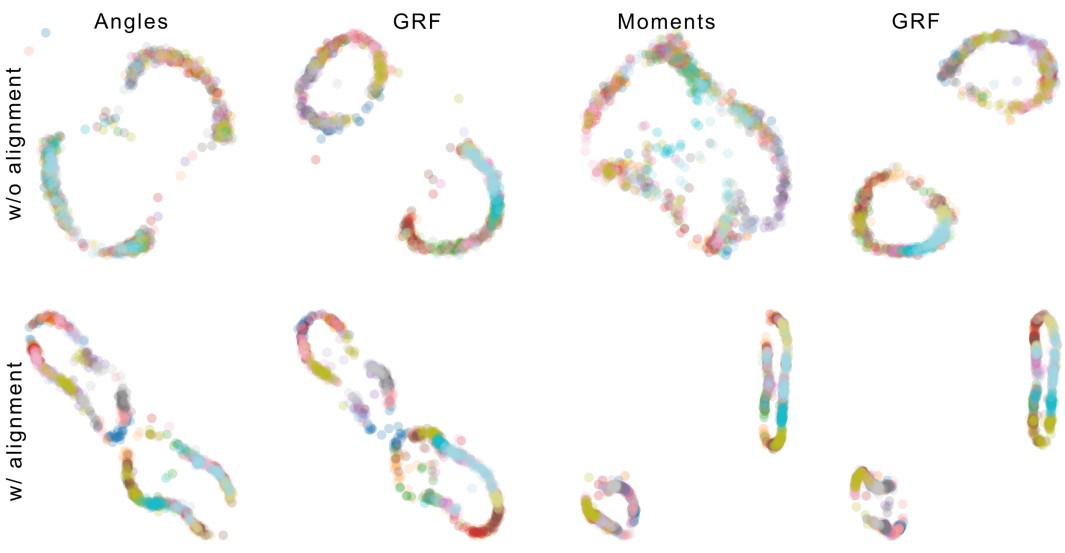

Figure 9: UMAP visualizations of latent spaces of cross-modal diffusion models for joint angles and GRF (left) and joint moments and GRF (right) for diffusion models trained independently (w/o alignment) and with latent alignment. The latent space of the latent aligned models shows a correlation in the structure and arrangement of locomotion tasks (color codes), whereas the latent space of the independently trained models shows a modality-specific structure without observable correlations.

Table 6: Comparison of cross-modal generation performance (quantified by MSE, mean $\pm$ std) of a diffusion model with shared encoder and modality-specific decoders with modality-specific encoder-decoder architecture trained with latent alignment.

| | Angles-moments | | Moments-GRF | | Angles-GRF | |
| --- | --- | --- | --- | --- | --- | --- |
| | $X\|Y$ | $Y\|X$ | $X\|Y$ | $Y\|X$ | $X\|Y$ | $Y\|X$ |
| Shared encoder | 0.98±0.02 | 1.02±0.03 | 1.00±0.02 | 1.13±0.04 | 0.95±0.01 | 1.14±0.03 |
| LaMbDA (ours) | 0.14±0.02 | 0.07±0.01 | 0.07±0.02 | 0.03±0.02 | 0.19±0.03 | 0.06±0.03 |

## A.6 FURTHER EXPERIMENTAL DETAILS

### A.6.1 DATASET

We used open-source biomechanical motion datasets Embry et al. (2018) consisting of locomotion data collected as multiple subjects walked on an instrumented treadmill at varying speeds (0.8 m/s, 1.0 m/s, and 1.2 m/s) and inclines (-10 °to 10 °at 2.5 °increments). The locomotion data was recorded using a 10-camera Vicon motion capture system, while the force plates in the treadmill recorded ground reaction forces (GRF). The processed data consists of three modalities 1) Kinematics that consists of 3D joint angles of hip, knee, and ankle, and 3D pelvis and foot angles, 2) joint kinetics that consists of 3D moments of hip, knee, and ankle, and 3) 3D ground reaction forces. The feature sets are represented as $(\theta_{hip,x}, \theta_{hip,y}, \theta_{hip,z}, \theta_{knee,x}, \theta_{knee,y}, \theta_{knee,z}, \theta_{ankle,x}, \theta_{ankle,y}, \theta_{ankle,z}, \theta_{foot,x}, \theta_{foot,y}, \theta_{foot,z}, \theta_{pelvis,x}, \theta_{pelvis,y}, \theta_{pelvis,z}), (\tau_{hip,x}, \tau_{hip,y}, \tau_{hip,z}, \tau_{knee,x}, \text{knee}, y, \tau_{knee,z}, \tau_{ankle,x}, \tau_{ankle,x}, \tau_{ankle,x}), (\text{GRF}_x, \text{GRF}_y, \text{GRF}_z)$. The features were normalized prior to model training.

### A.6.2 MODEL ARCHITECTURE

We trained parallel diffusion models (DDPM), $p_\theta(\mathbf{X}|\mathbf{Y})$ and $p_\phi(\mathbf{Y}|\mathbf{X})$ for generating the two modalities $\mathbf{X}$ and $\mathbf{Y}$ conditioned on the other with latent alignment. Each model has the same architecture and consists of four modules: 1) an input encoder, that encodes the noise input, designed as a transformer-based encoder with four layers and a model dimension of 128, 2) a condition embedder, which encodes the guiding signal, 3) a timestep embedder, that encodes the diffusion timestep $t$, designed as a multilayer perceptron (MLP) with SiLU Wang et al. (2018) activation, and 4) an output decoder, that generates the output at each diffusion timestep, designed as a transformer decoder with four layers. At each diffusion timestep, the noise input is linearly projected from the input space to the model space and combined with a positional and time embedding, before it passes through the encoder. At the decoder, cross-attention is computed between the condition embedding combined with positional and time embedding and the encoded noise input. The generated output is linearly projected onto the output space. Each model has ˜25M tunable parameters.

### A.6.3 IMPLEMENTATION DETAILS

We trained parallel diffusion models (DDPM), $p_\theta(\mathbf{X}|\mathbf{Y})$ and $p_\phi(\mathbf{Y}|\mathbf{X})$ for generating the two modalities $\mathbf{X}$ and $\mathbf{Y}$ conditioned on the other with latent alignment. Each model has the same architecture and consists of a transformer encoder and decoder, each with four layers. Inputs to both encoder and decoder were combined with sinusoidal position encoding and time embedding. The decoder additionally takes a conditional embedding derived from the other modality through a linear layer. Each model has ˜25M tunable parameters. In contrast to having a single test set, we performed a K-fold cross-validation of the models by creating multiple versions of train and test datasets. The model training was done for ˜50 epochs on an RTX4090 GPU computer which consumed ˜10GB of GPU space and ˜10 hours for 10 cross-validation iterations.

### A.6.4 METRICS

**Mean-Squared Error (MSE)** Each diffusion model, $p(\mathbf{X}|\mathbf{Y})$ or $p(\mathbf{Y}|\mathbf{X})$, generates data for one modality, $\hat{\mathbf{X}}$ or $\hat{\mathbf{Y}}$, conditioned on the other. Since the modalities are time-series data that correspond to each other, this conditioned generation can be viewed as a cross-modal reconstruction task. The ground truth signal for the reconstructed data is defined as the temporal counterpart of the conditioning

data. We then calculate the mean squared error (MSE) between the generated data, $\hat{\mathbf{X}}$ or $\hat{\mathbf{Y}}$ and the ground truth data for the respective modality.

**Fréchet Inception Distance (FID)**    evaluates the quality of generated data by measuring the Fréchet distance (Wasserstein-2 distance) between the distributions of real and generated features Yu et al. (2021). Originally designed for images, we adapt this metric for generated time series data by computing the distance in the temporal space. Given two Gaussian distributions, $\mathcal{N}(\mu, \Sigma)$ and $\mathcal{N}(\mu', \Sigma')$, respectively fitted to the real and generated feature representations, the FID is computed as:

$$\text{FID} = \|\mu - \mu'\|_2^2 + \text{tr}(\Sigma + \Sigma' - 2(\Sigma\Sigma')^{\frac{1}{2}}) \tag{10}$$

**Predictive score**    This metric evaluates generation quality by assessing how well a model trained on generated data predicts future values in real data Yoon et al. (2019). A sequence-to-sequence model (e.g., LSTM) is trained to predict the latter part of a time series from its initial part, and its performance on real data reflects the quality of the generated data, with lower errors indicating higher quality.

### A.6.5 REPRESENTATION ALIGNMENT METHODS

**SimCLR**    Chen et al. (2020b) is a contrastive learning approach that learns representations by bringing similar samples (positive pairs) closer in the latent space while pushing dissimilar ones (negative pairs) apart. It relies on a contrastive loss function, the Normalized Temperature-scaled Cross-Entropy Loss (NT-Xent loss), which is defined as:

$$\ell_{i,j} = -\log \frac{\exp\left(\text{sim}(\mathbf{z}_i, \mathbf{z}_j)/\tau\right)}{\sum_{k=1}^{2N} 1_{[k \neq i]} \exp\left(\text{sim}(\mathbf{z}_i, \mathbf{z}_k)/\tau\right)} \tag{11}$$

where $\mathbf{z}_i, \mathbf{z}_j$ are the embeddings of two samples, $\text{sim}(\mathbf{z}_i, \mathbf{z}_j) = \frac{\mathbf{z}_i \cdot \mathbf{z}_j}{\|\mathbf{z}_i\|\|\mathbf{z}_j\|}$ is the cosine similarity measure, $\tau$ is the temperature scaling parameter, and $N$ is the batch size. The total loss across a batch of size $N$ is computed as:

$$\mathcal{L}_{\text{SimCLR}} = \frac{1}{2N} \sum_{i=1}^{N} \left(\ell_{2i-1,2i} + \ell_{2i,2i-1}\right). \tag{12}$$

We consider the latent embeddings of the corresponding samples of both modalities in a batch as positive pairs, and non-corresponding samples as negative pairs.

**Barlow Twins**    Zbontar et al. (2021) addresses the limitations of contrastive methods by eliminating the need for negative samples. It introduces a loss function that aligns the cross-correlation matrix of embeddings from two identical networks processing different augmentations of the same image (in our case two modalities). The objective is twofold: (1) to make the diagonal elements of this matrix approach one, ensuring invariance, and (2) to drive the off-diagonal elements towards zero, promoting redundancy reduction. This strategy effectively prevents collapse by decorrelating different dimensions of the representation space.

Given two embeddings $\mathbf{z}^A$ and $\mathbf{z}^B$ (where $A$ and $B$ are two modalities), it computes the cross-correlation matrix:

$$C_{ij} = \frac{1}{B} \sum_{n=1}^{N} z_n^A(i) z_n^B(j) \tag{13}$$

where $N$ is the batch size and $z^{(\cdot)}(i)$ represents the $i$-th feature dimension. The Barlow Twins loss consists of two key terms:

- **Invariance term**: Ensures that representations of the same input under different augmentations are similar: $\sum_i (1 - C_{ii})^2$.

- **Redundancy reduction term**: Enforces decorrelation across different dimensions to prevent representational collapse: $\sum_{i \neq j} C_{ij}^2$.

The final loss function is formulated as:

$$\mathcal{L}_{\text{Barlow}} = \sum_i (1 - C_{ii})^2 + \lambda \sum_{i \neq j} C_{ij}^2, \tag{14}$$

where $\lambda$ is a balancing hyperparameter.

**VICReg** (Variance-Invariance-Covariance Regularization) Bardes et al. (2021) extends Barlow Twins by adding an explicit variance regularization term, preventing representational collapse through three objectives:

- **Invariance**: Ensures consistency between augmented views, similar to SimCLR and Barlow Twins: $\mathcal{L}_{\text{inv}} = \sum_{i=1}^{d} \|\mathbf{z}^A(i) - \mathbf{z}^B(i)\|^2$ where $A$ and $B$ are two modalities, and $z^{(\cdot)}(i)$ represents the $i$-th feature dimension.

- **Variance regularization**: Ensures that the standard deviation of each embedding dimension $i$ remains above a threshold $\gamma$, preventing collapse to trivial solutions: $\mathcal{L}_{\text{var}} = \sum_{i=1}^{d} \max(0, \gamma - \sigma(\mathbf{z}(i)))^2$.

- **Covariance regularization**: Reduces redundancy between different dimensions by minimizing off-diagonal terms of the covariance matrix: $\mathcal{L}_{\text{cov}} = \sum_{i \neq j} C_{ij}^2$, $\quad C = \frac{Z^\top Z}{N}$, where $N$ is the batch size.

The total VICReg loss function is:

$$\mathcal{L}_{\text{VICReg}} = \lambda_{\text{inv}} \mathcal{L}_{\text{inv}} + \lambda_{\text{var}} \mathcal{L}_{\text{var}} + \lambda_{\text{cov}} \mathcal{L}_{\text{cov}}. \tag{15}$$

This approach provides a balance between alignment and diversity constraints, ensuring that representations are meaningful, discriminative, and well-distributed.

## A.7 HYPERPARAMETER SENSITIVITY ANALYSIS

To assess the robustness of LaMbDA with respect to key modeling choices, we conducted sensitivity analyses on two primary hyperparameters: the latent dimensionality $D$ of the diffusion encoders and the sequence length $L$ used for training.

**Latent dimensionality.** We evaluated LaMbDA using latent dimensions $D \in \{32, 64, 128, 256\}$, keeping sequence length $L = 300$. As shown below for the Moments–GRF pair (Tab. 7, performance remains stable across a wide range of latent sizes, with improvements with model dimension $D$ until our choice of $D = 128$.

**Sequence length.** We further assessed sensitivity to sequence length using $L \in \{150, 300, 450\}$ keeping the model dimension $D = 128$. Since a single gait cycle consists of approximately 150 samples, this range spans single-cycle and multi-cycle contexts. We found that performance improves with sequence length until $L = 300$, after which it saturates.

## A.8 ENTROPY ANALYSIS

In table 1 of the paper, we observed asymmetries in the reconstruction metrics for $X \mid Y$ and $Y \mid X$. We hypothesized that this arises from the asymmetry in the information content in $X$ and $Y$ modalities about each other. Joint angles provide a richer and more complete description of the underlying movement state than the downstream kinetic measurements. Consequently, reconstructing

| $X$ = **Moments**, $Y$ = **GRF** | **Model dimension,** $D$ ($L = 300$) | | | | **Sequence length,** $L$ ($D = 128$) | | |
|---|---|---|---|---|---|---|---|
| | $D = 32$ | $D = 64$ | $D = 128$ | $D = 256$ | $L = 150$ | $L = 300$ | $L = 450$ |
| $X \mid Y$ | 0.045 | 0.039 | 0.037 | 0.038 | 0.044 | 0.037 | 0.037 |
| $Y \mid X$ | 0.020 | 0.021 | 0.014 | 0.013 | 0.017 | 0.014 | 0.014 |

Table 7: Results of hyperparameter sensitivity analysis. We analyzed the effect of model dimension, $D$, and sequence length, $L$, on the reconstruction accuracy.

| | $H(X \mid Y)$ | $H(Y \mid X)$ |
|---|---|---|
| $X$ = Angles, $Y$ = Moments | $2.12 \pm 0.56$ | $-3.55 \pm 0.40$ |
| $X$ = Angles, $Y$ = GRF | $2.38 \pm 0.54$ | $-6.74 \pm 0.44$ |
| $X$ = Moments, $Y$ = GRF | $-3.52 \pm 0.45$ | $-6.96 \pm 0.45$ |

Table 8: Conditional entropies $H(X \mid Y)$ and $H(Y \mid X)$ for angles-moments, angles-GRF, and moments-GRF pairs.

$Y$ from $X$ is an inherently easier task with lower uncertainty. In contrast, reconstructing angles (X) from moments ($Y$) is a harder problem: the kinetic signals contain less information about the full kinematic trajectory.

To quantify this effect, we computed conditional entropies $H(X \mid Y)$ and $H(Y \mid X)$ for the biomechanical modality pairs. We used k-nearest neighbor (kNN) estimators with k=20 to estimate the mutual information I(X; Y) using Kraskov–Stögbauer–Grassberger, KSG estimator, and differential entropy $H(X)$ using Kozachenko–Leonenko estimator for continuous variables $X$ and $Y$ (and hence the negative values of entropy). The conditional entropy is computed as $H(X \mid Y) = H(X) - I(X;Y)$ and $H(Y \mid X) = H(Y) - I(X;Y)$. As expected, we find that the uncertainty of angles given moments is substantially higher than the uncertainty of moments given angles.

These results also explain why, in the ablation experiments (Tab. 5), only modest improvements were obtained for $Y \mid X$ with the addition of more LaMbDA loss components. Since moments given angles ($X \mid Y$) are the easier direction, it is less sensitive to ablations, while angles given moments ($Y \mid X$) are a harder task and benefit more strongly from the full LaMbDA objective.

