# OpenReview forum: "LaMbDA: Local Latent Embedding Alignment for Cross-modal Time-Series Diffusion"
_ICLR.cc/2026/Conference — Submitted to ICLR 2026_

### Official Review · Reviewer_XiJt · 2025-10-30

**Soundness:** 2
**Presentation:** 3
**Contribution:** 2
**Rating:** 4
**Confidence:** 3

**Summary:**

The paper proposes a diffusion-based latent manifold alignment method to achieve two-modality learning for dynamical systems.

**Strengths:**

The paper is well-structured, and the model is clearly presented. The theoretical foundations are included. Experiments demonstrate the mechanism and the superior performance.

**Weaknesses:**

The baseline comparisons are missing. Some analysis needs to be added.

**Questions:**

1.	In table 4, can the author explain why the result of Y|X is similar for different ablation scenarios?
2.	The author is suggested to give some raw trajectory visualization for X and Y to show their synchronized behaviors, which motivates the alignment loss.
3.	The baseline of diffusion model, dynamic prediction models, and other time-series methods is lacking. For example, other methods may perform well in the reconstruction of Fig. 2B.
4.	The sensitivity analysis with respect to the hyperparameters is lacking.
5.	The limitations are absent.

---

> ### Author Response · Authors · 2025-11-21
> **Rebuttal part 1**
>
> We thank the reviewer for the positive assessment of the paper’s structure, theoretical grounding, and empirical evaluation. We also appreciate the constructive suggestions for strengthening the work. Below, we address each comment point by point.
>
> **Baseline comparisons**: We thank the reviewer for highlighting the need for stronger baselines. In response, we have substantially expanded our experimental comparisons. In addition to the self-supervised alignment baselines already included in Table 3, we now compare LaMbDA against state-of-the-art cross-modal diffusion frameworks such as CDCD CDCD (Zhu et al., 2022) and CMMD (Yang et al., 2024), as well as a conditional generative baseline, namely a Conditional VAE (with and without latent alignment), and a simple transformer-based time-series regressor.
>
> ||Linear probing||Nonlinear probing||
> |-|-|-|-|-|
> |X=Angles,Y=Moments|X\|Y|Y\|X|X\|Y|Y\|X|
> |Transformer Regressor|0.05|0.05|0.05|0.05|
> |CVAE w/o align|0.48|0.35|0.65|0.4|
> |CVAE w/align|0.54|0.53|0.69|0.67|
> |CDCD|0.06|0.47|0.04|0.5|
> |CMMD|0.65|0.78|0.76|**0.89**|
> |Ours|**0.89**|**0.88**|**0.89**|0.88|
>
> Across all these baselines, LaMbDA consistently achieves the strongest performance. For instance, in the angles–moments setting, CMMD achieves competitive nonlinear probing scores, but LaMbDA matches or exceeds it in both directions (e.g., X|Y: 0.89 vs. 0.76; Y|X: 0.88 vs. 0.89). The transformer regressor and CVAE variants perform substantially worse, with the CVAE improving somewhat when alignment is added but still falling short of diffusion-based latent alignment. We will add these expanded baseline comparisons and corresponding tables to the manuscript.
>
> **Explanation of ablation results**: We thank the reviewer for this insightful observation. The fact that the Y|X results remain similar across ablation settings is likely related to the information asymmetry between the modalities. In Table 4, X denotes joint angles (15 dimensions) and Y denotes joint moments (9 dimensions), as described in Appendix A.6.1. Joint angles provide a richer and more complete description of the underlying movement state than the downstream kinetic measurements. Consequently, reconstructing  Y from X is an inherently easier task with lower uncertainty. In such an easier reconstruction regime, adding more components of the LaMbDA loss provides only modest performance improvement, leading to similar Y|X values observed across ablations.
>
> In contrast, reconstructing angles (X) from moments (Y) is a harder problem: the kinetic signals contain less information about the full kinematic trajectory. In this more challenging setting, latent alignment plays a much more significant role, and Table 4 shows that each component of the LaMbDA loss contributes meaningfully to improving X|Y reconstruction.
>
> To quantify this effect, we computed conditional entropies H(X∣Y) and H(Y∣X) for the Angle–Moment pair. As expected, we find that the uncertainty of angles given moments is substantially higher than the uncertainty of moments given angles:
> ||H(X\|Y)|H(Y\|X)|
> |-|-|-|
> |X=Angles,Y=Moments|2.12$\pm$0.56|-3.55$\pm$0.40|
> |X=Angles,Y=GRF|2.38$\pm$0.54| -6.74 $\pm$ 0.44 |
> |X=Moments, Y=GRF|-3.52$\pm$0.45|-6.96$\pm$0.45|
>
> These values empirically support the intuition that moments given angles (X|Y) is the easier direction, and therefore less sensitive to ablations, while angles given moments (Y|X) benefits more strongly from the full LaMbDA objective. We will incorporate this explanation into the revised manuscript.
>
> **Trajectory visualizations**: We thank the reviewer for this helpful suggestion. Appendix Fig. 6 visualizes real and reconstructed trajectories for each modality separately (angles in the top panel and moments in the bottom panel), but we agree that including raw, time-aligned X–Y segments (e.g., angles–moments or angles–GRF) would more clearly illustrate the temporal synchrony that motivates our alignment loss. In the revised manuscript, we will add these synchronized multimodal visualizations to make this relationship explicit.

---

> ### Author Response · Authors · 2025-11-21
> **Rebuttal part 2**
>
> Continuation from Rebuttal Part 1:
>
> **Hyperparameter sensitivity analysis**: We thank the reviewer for highlighting this point. We have begun a sensitivity analysis for the most important hyperparameters in our framework: latent dimension $D$ and sequence length $L$. Focusing first on the latent dimensionality using a grid of {32,64,128}, our preliminary results (shown below for a subset of the data for the Moments–GRF pair) indicate that LaMbDA is robust across a broad range of latent sizes, with performance improving with increasing latent dimension:
> |X=Moments,Y=GRF|D=32|D=64|D=128|
> |-|-|-|-|
> |X\|Y|0.045|0.039|0.037|
> |Y\|X|0.020|0.021|0.014|
>
> We are also running a sequence-length sensitivity analysis over the grid {50,150,300,450}. Since a single gait cycle consists of 150 samples, this range allows us to assess performance on sub-cycle, single-cycle, and multi-cycle windows. We will include the full results for both latent dimensionality and sequence length in the revised manuscript once experiments are complete.
>
> **Limitations**: We thank the reviewer for this comment. As noted briefly in the Conclusion and Future Work section, our approach has two main limitations that we will make more explicit in the revised manuscript. First, LaMbDA assumes that the paired modalities arise from a shared latent dynamical process. Extending the method to settings where this assumption is violated or only partially satisfied is an important direction for future work. Second, while our formulation is naturally bidirectional for two modalities, scaling to three or more modalities may require additional strategies (e.g., centroid-based alignment or coordinated pairwise alignment). We will add a dedicated limitations section to clearly outline these points.

---

> > ### Author Response · Authors · 2025-11-30
> > **Hyperparameter sensitivity analysis**
> >
> > We have now finished the hyperparameter sensitivity analysis for our approach. As mentioned earlier, we conducted a sensitivity analysis for the most important hyperparameters in our framework: latent dimension $D$ and sequence length $L$.
> >
> > **Latent dimensionality.** We evaluated LaMbDA using latent dimensions $D\in\{32,64,128,256\}$, keeping sequence length $L = 300$. The performance remains stable across a wide range of latent sizes, with improvements with model dimension $D$ until our choice of $D=128$.
> >
> > | X=Moments, Y=GRF | D=32 | D=64 | D=128 | D=256 |
> > |:---:|:---:|:---:|:---:|:---:|
> > | X\|Y | 0.045 | 0.039 | 0.037 | 0.038 |
> > | Y\|X | 0.020 | 0.021 | 0.014 | 0.013 |
> >
> > **Sequence length.** We further assessed sensitivity to sequence length using $L\in \{150,300,450\} $ keeping the model dimension $D=128$. Since a single gait cycle consists of approximately 150 samples, this range spans single-cycle, and multi-cycle contexts. We found that performance improves with sequence length until $L=300$, after which it saturates.
> >
> > | X=Moments, Y=GRF | L=150 | L=300 | L=450 |
> > |:---:|:---:|:---:|:---:|
> > | X\|Y | 0.044 | 0.037 | 0.037 |
> > | Y\|X | 0.017 | 0.014 | 0.014 |
> >
> > We will include the results of the hyperparameter sensitivity analysis to the revised manuscript.

---

### Official Review · Reviewer_PVqD · 2025-10-30

**Soundness:** 2
**Presentation:** 3
**Contribution:** 2
**Rating:** 6
**Confidence:** 3

**Summary:**

This paper proposed a difussion framework for cross-modal latent alighment called LaMbDA. The framework used one order and two order loss for reguliraztion of the model training. The model was trained and tested on synsetic Lorenz system data, and locomotor data. The model experiment showed the alignment with MSE and FID, compared with self-supervised learning, showed the learned representation using UMAP, did ablation study on terms in the loss function. The figures and tables support their claim that the alignment increases the generation fidelity.

**Strengths:**

The paper gave a novel frame work with two identical diffusion for the latent alignment. I have checked the formulas in Section 3, they all make sense and does not have ambiguity or miss leading notation. The paper did plenty experiments to support its claim, from the values of the figures and tables. All the details of the implementation I could found described in the Appendix.

**Weaknesses:**

The experiment with the alignment of angles-moments, angles-GRF, moments-GRF, seems all 3-D vectors if I understand correctly. If like that, I would like to see the author could align diffrent high dimensional observations.

**Questions:**

1. Have you tried to see the results of different latent dimension?

2. For some of the pairs, angles-moments, angles-GRF, moments-GRF,  the difference of the results of X on Y and Y on X is large, for example in Table 1 (0.14 vs 0.07, 0.19 vs 0.06, 0.07 vs 0.03 for MSE, same big difference for FID) , have you considered the reason why this happens?

3. Do you have an order of sampling X and Y within time step and the first step, does this involve bias and relate to the values of X conditioned on Y and Y conditioned on X?

---

> ### Author Response · Authors · 2025-11-21
> **Rebuttal**
>
> We thank the reviewer for their thoughtful assessment and positive evaluation of the novelty of our study, the clarity of the notation, and the thoroughness of the experimental and implementation details and empirical evaluation. We address the remaining questions and concerns point by point below.
>
> **Dimensionality of observations**: Thank you for raising this important point. The three biomechanical modalities we align, joint angles, joint moments, and ground-reaction forces, indeed differ substantially in dimensionality. As mentioned in Appendix A.6.1, angles have 15 dimensions ($\theta_{hip}^{x,y,z}, \theta_{knee}^{x,y,z}, \theta_{ankle}^{x,y,z}, \theta_{foot}^{x,y,z}, \theta_{pelvis}^{x,y,z}$), moments 9 ($\tau_{hip}^{x,y,z}, \tau_{knee}^{x,y,z}, \tau_{ankle}^{x,y,z}$), and GRFs 3 ($GRF^{x,y,z}$). LaMbDA can align these modalities with different dimensionalities as the alignment operates in the latent space of the diffusion encoders. The results in Table 1 demonstrate that local latent alignment remains effective across these heterogeneous observation spaces.
>
> We agree that this distinction should be made explicit earlier in the paper and will move the modality dimensionalities from the appendix into the main text.
>
> **Different latent dimensions**: Thank you for this excellent suggestion. We have now conducted a hyperparameter sensitivity analysis to examine how the choice of latent dimensionality influences cross-modal generation performance. Preliminary results of reconstruction MSE on a subset of data for the moments–GRF pair are shown below:
> |X=Moments,Y=GRF|D=32|D=64|D=128|
> |-|-|-|-|
> |X\|Y|0.045|0.039|0.037|
> |Y\|X|0.020|0.021|0.014|
>
> These findings suggest that LaMbDA performs reliably across latent sizes, with performance improving gradually as dimensionality increases. We are running the full set of experiments and will include the complete results in the revised manuscript.
>
> **Difference in metrics for X and Y** Thank you for this thoughtful observation. The asymmetry between $X∣Y$ and $Y∣X$ likely arises from the intrinsic biomechanical information content of the biomechanical modalities. As we mentioned in our previous response, the modalities differ substantially in dimensionality: angles have 15 dimensions, moments 9, and GRFs 3. However, the key source of asymmetry is not dimensionality alone but the underlying biomechanics. Joint angles describe the configuration and motion of the limb segments, which provide a richer representation of the movement state than the downstream kinetic signals. Joint moments depend on these kinematics through inverse dynamics, and GRFs depend on the global body motion and foot–ground interaction, which are also constrained by the kinematic trajectory. As a result, the angle modality typically contains more upstream information about the ongoing movement than moments or GRFs. This makes reconstructing moments or GRFs conditioned on angles easier than reconstructing the full kinematic trajectory from the lower-dimensional kinetic measurements. This explains the directional differences observed in the reconstruction errors in Table 1.
>
> To quantify this systematically, we estimated the conditional entropies, H(X∣Y) and H(Y∣X). Across all modality pairs, the effective uncertainty of angles given moments or GRFs is markedly higher than the uncertainty of moments or GRFs given angles:
> | | H(X\|Y) | H(Y\|X) |
> |-|-|-|
> |X=Angles,Y=Moments|2.12$\pm$0.56|-3.55$\pm$0.4|
> |X=Angles,Y=GRF|2.38$\pm$0.54 | -6.74 $\pm$ 0.44|
> |X=Moments,Y=GRF|-3.52$\pm$0.45|-6.96$\pm$0.45|
>
> These values reflect the expected information asymmetry (differential entropies can be negative), supporting why the reconstruction errors differ in the two directions. We will incorporate this explanation and the supporting entropy analysis into the revised manuscript.
>
> **Order of sampling of X and Y** Thank you for this question. We interpret it as referring to whether the two conditional models $p_\theta(X∣Y)$ and $p_\phi(Y|X)$ are evaluated sequentially within each diffusion timestep during training, and whether such an ordering could bias the results. In our implementation, both models are evaluated independently and symmetrically at each diffusion timestep t. The prediction of X does not depend on the prediction of Y, nor vice versa; the models $p_\theta(X|Y)$ and  $p_\phi(Y|X)$ use their own noisy inputs $X_t$ and $Y_t$ and their corresponding conditioning signals $Y$ and $X$ respectively. Their losses are then combined, and the parameters $\theta$ and $\phi$ are updated jointly in the same optimizer step. Thus, there is no sequential computation or ordering bias in the training procedure.
>
> At inference time, sampling $X∣Y$ and $Y∣X$ are likewise independent processes, each conditioned on the real observed modality. Therefore, the directional asymmetries reported in Table 1 do not arise from sampling order, but from the intrinsic information asymmetry between the modalities, as discussed above.

---

### Official Review · Reviewer_6MXH · 2025-11-03

**Soundness:** 2
**Presentation:** 3
**Contribution:** 2
**Rating:** 4
**Confidence:** 4

**Summary:**

In this work, the authors proposed a multi-modal alignment framework named LAMbDA for cross-modal time-series data. Given the two paired observed modalities X and Y from a shared latent dynamical process, the main idea of the paper is to add the alignment regularizors on the latent hidden states of these two modalities data to add the inductive biases constraints to the model. LaMbDA adds this alignment using a combination of a first-order sequence-contrastive loss and a second-order covariance alignment term. On the experimental side, the authors evaluate the effectiveness of the proposed method on synthesized Lorenz dynamics dataset and the human biomechanical locomotor datasets.

**Strengths:**

1. The paper is well-written and easy for the audiance to read and follow.
2. The modeling of observed multi-modal data and its related alignment tasks are critical problems in dynamical systems.
3. The proposed LAMbDA framework significantly improves cross-modal generation performance compared to the non-aligned baseline across multiple modality pairs and metrics (e.g., MSE, FID, Predictive score).

**Weaknesses:**

1. I think that the main algorithm novelty of this work LaMbDA mainly comes from the combination of a not new contrastive loss term [1] and a covariance loss (second-order), it's actually a bit empirical. The difference is only that this paper puts them onto this new dynamical system alignment application context.
2. While focused in dynamical systems scenarios, the proposed alignment method does not explicitly enforce alignment based on the dynamics or flow of the system (e.g., considering vector fields or preserving local structures beyond the simple Taylor approximation).
3. In the experiments, the paper only compares the generative performance against non-aligned models. And also some baselines, like MSE and SimCLR, are actually weak, To be considered as a performant method, it must benchmark LAMbDA's synthesis quality against other powerful generative architectures with alignment.

[1] Representation Learning with Contrastive Predictive Coding. 2018. Oord, et al.

**Questions:**

I have no more questions, other concerns please relate to my weaknesses section.

---

> ### Author Response · Authors · 2025-11-21
> **Rebuttal**
>
> We thank the reviewer for the constructive feedback and for recognizing the significance of the problem setting as well as the empirical gains demonstrated by our approach. Below, we address each of the comments.
>
> **Novelty**: Thank you so much for giving us the opportunity to clarify this further. I think it would have been more helpful if we would have clarified the difference between the LaMbDA  loss, and the LaMbDA  framework better. LaMbDA loss builds on established contrastive objectives for doing the alignment, but the paper`s main contribution lies in the LaMbDA  framework. where, we introduce step-wise, local latent alignment (using the LaMbDA  loss) at every denoising step of a bidirectional cross-modal diffusion model, creating an interaction between alignment and diffusion dynamics that. This, to our knowledge, has not been explored before.
>
> At each timestep, both models produce time-matched latent subsequences. The LaMbDA loss aligns these windows using first-order contrastive similarity and second-order covariance structure, applied directly within the reverse-diffusion trajectory rather than as an external encoder regularizer. Repeating this at all T diffusion steps yields a consistently coupled latent manifold whose dynamics are consistent across both modalities and across noise levels.
>
> This design is motivated directly by a dynamical-systems view. Motivated by Takens-style delay embeddings, each modality yields a smooth reconstruction of the same latent dynamical state. Aligning first- and second-order local structure at each timestep is therefore a way of ensuring that both modalities encode compatible local phase-space neighborhoods that leverages the structure of shared underlying dynamics and operates across noise levels.
>
> Thanks again. We will clarify this in the manuscript. I hope our explanation now makes the novelty a bit clearer (?) Please let us know if you have any questions and we are more than happy to clarify further.
>
> **Alignment based on flow or vector field**: We thank the reviewer for this insightful comment. Rather than modeling the vector field or flow explicitly, our approach relies on Takens-style delay coordinate embeddings to incorporate the flow: a sufficiently rich delay embedding reconstructs a diffeomorphic image of the underlying dynamical system, and the local geometry of this reconstruction already encodes the system’s flow. Each modality, therefore, provides a smooth, partial reconstruction of the same latent dynamics, and the first-order contrastive and second-order covariance terms in LaMbDA align these local neighborhoods at every diffusion step.
>
> Because the vector field is implicit in the manifold’s local geometry, aligning the embedded manifolds themselves effectively aligns the dynamical structure without requiring an explicit flow parameterization or derivative supervision.
>
> **Comparison against benchmarks**: We appreciate the reviewer’s suggestion that stronger baselines would further clarify the contribution of LaMbDA. In addition to SimCLR, Barlow Twins, VICReg, and MSE alignment, we now include comparisons against state-of-the-art cross-modal diffusion frameworks, specifically CDCD (Zhu et al., 2022) and CMMD (Yang et al., 2024), as well as conditional generation baselines including a CVAE (with and without alignment) and a simple transformer-based regressor.
> |  | Linear probing |  | Nonlinear probing |  |
> |-------------------------|----------------|----------|-------------------|----------|
> | X = Angles, Y = Moments | X\|Y | Y\|X | X\|Y | Y\|X |
> | Transformer Regressor   | 0.05  | 0.05 | 0.05  | 0.05 |
> | CVAE w/o align | 0.48  | 0.35  | 0.65 | 0.4 |
> | CVAE w/align | 0.54 | 0.53 | 0.69 | 0.67 |
> | CDCD  | 0.06 | 0.47 | 0.04 | 0.5 |
> | CMMD  | 0.65 | 0.78 | 0.76 | **0.89** |
> | Ours | **0.89** | **0.88** | **0.89** | 0.88 |
>
> Across these additional baselines, LaMbDA consistently achieves the strongest performance. In the angles–moments setting, for instance, CMMD reaches competitive nonlinear probing scores, yet LaMbDA matches or exceeds it in both directions (e.g., X|Y: 0.89 vs. 0.76; Y|X: 0.88 vs. 0.89). The deterministic transformer regressor and CVAE baselines perform substantially worse overall. While adding latent alignment improves CVAE performance, it still trails behind the diffusion-based alignment achieved by LaMbDA. We will include these additional results in the revised manuscript. Overall, LaMbDA offers clear improvements over both state-of-the-art alignment methods and modern cross-modal diffusion architectures.
>
> **References**
>
> Zhu, Y., Wu, Y., Olszewski, K., Ren, J., Tulyakov, S., & Yan, Y. (2022). Discrete contrastive diffusion for cross-modal music and image generation. arXiv preprint arXiv:2206.07771.
>
> Yang, R., Gamper, H., & Braun, S. (2024, September). Cmmd: Contrastive multi-modal diffusion for video-audio conditional modeling. In European Conference on Computer Vision (pp. 214-226). Cham: Springer Nature Switzerland.

---

### Meta-Review · Area_Chair_DJY8 · 2026-01-06

**Summary:**

This work aims to create a multi-modal shared latent state through coupled autoencoders. The goal of this process is to be able to infer time series from each other. The results are demonstrated on both real data (locomotion dataset) and several synthetic datasets. There were several concerns stated and unfortunately the reviewers did not respond in time to the author rebuttal. Given those concerns and what I can read from the response, the main issue is the fairness/extent of baseline assessment and comparisons. There were a few additional baselines added after the discussion, however there is extensive literature on shared subspaces in multi-view data that was largely ignored and would have validated the use of dynamics as part of the model. This makes the paper feel incomplete. I've included a handful of the citations below, some of which also include the ideas of manifold-style regularization in the latent space alignment and directional prediction via cross-encoders. Given this I feel that a more careful inclusion of the past literature would greatly improve a future submission.

- Lyu, et al. Understanding latent correlation-based multiview learning and self-supervision: An identifiability perspective. 2021
- Gondur et al. Multimodal gaussian process variational autoencoders for neural and behavioral data. ICLR, 2023.
- Karakasis & Sidiropoulos. Revisiting deep generalized canonical correlation analysis. arXiv, 2023.
- Kevrekidis et al. Conformal disentanglement: A neural framework for perspective synthesis and differentiation. arXiv, 2024
- Koukuntla et al. Unsupervised discovery of the shared and private geometry in multi-view data. arxiv 2024

**Reviewer Concerns:**

While there remains room to improve, the authors did address a good number of concerns (from what I can tell and to the extent that I can validate) including
 - extending the dimensionality of the data to demonstrate that the model works in higher dimensions,
 - Adding sensitiveity analyses to the length of the sequences and number of dimensions
 - Adding additional baselines
 - Clarification on the assymetry of the results in predicting in a directional way

I do think that the sensitivity analysis is still weak in that only 2 parameters were tested, and that the literature is not adequately compared to given all the work in multi-view data.

**Reviewer Scores:**

The initial scores for this paper were 4,4,6. The authors did do additional work but I feel that the comparisons are still lacking and the sensitivity analysis only spans two parameters that might not convince the reviewer. Unfortunately this submission suffered from none of the reviewers responding, as well as 3 fairly lackluster reviews. I would guess that at best the scores might move to 4,6,6.

---

### Decision · Program_Chairs · 2026-01-26

Reject